# Degradation of *N*-(*n*-butyl) Thiophosphoric Triamide (NBPT) with and without Nitrification Inhibitor in Soils

**Ahmed A. Lasisi \*** and **Olalekan O. Akinremi**

Department of Soil Science, University of Manitoba, Winnipeg, MB R3T 5H3, Canada;
wole.akinremi@umanitoba.ca
\* Correspondence: greatahmlas@gmail.com

**Abstract:** Recent studies have shown that nitrification inhibitor (NI) impairs the efficacy of urease inhibitor, *N*-(*n*-butyl) thiophosphoric triamide (NBPT), in reducing ammonia volatilization and urea hydrolysis rate. A laboratory study was conducted to evaluate the influence of NI (specifically 3,4-dimethyl pyrazole phosphate) on the degradation of NBPT in six soils. Soils were amended with either NBPT (10 mg NBPT kg$^{-1}$ soil) or NBPT plus NI (DI; 10 mg NBPT + 2.5 mg NI kg$^{-1}$ soil), incubated at 21 °C, and destructively sampled eight times during a 14-day incubation period. The degradation of NBPT in soil was quantified by measuring NBPT concentration with high-performance liquid chromatography-mass spectrometry, and the degradation rate constant was modeled with an exponential decay function. The study showed that the persistence of NBPT in soil was not influenced by the presence of NI, as the NBPT degradation rate constant across soils was 0.5 d$^{-1}$ with either NBPT or DI. In contrast, the degradation rate constant was significantly dependent on soils, with values ranging from 0.4 to 1.7 d$^{-1}$. Soil pH was the most important variable affecting the persistence of NBPT in soils. The half-life of NBPT was 0.4 d in acidic soil and 1.3 to 2.1 d in neutral to alkaline soils. The faster degradation of NBPT in acidic soils may explain its reduced efficacy in such soils.

**Keywords:** NBPT; nitrification inhibitor; half-life; degradation rate constant

## 1. Introduction

Globally, urea is the predominant form of granular nitrogen (N) fertilizer used to supplement soil N availability to crops. It is relatively safe to handle and contains high N content (46%). When urea is applied to soils, it becomes hydrolyzed in the presence of the urease enzyme into one bicarbonate ion and two molecules of ammonia [1]. The process of urea hydrolysis increases the soil pH around urea, which drives the equilibrium between ammonia and ammonium toward ammonia, thereby resulting in the volatilization of ammonia. Ammonia volatilization from urea when it is surface-applied without incorporation could be greater than 20% of applied N and is one of the reasons for low urea-N use efficiency [2]. Apart from being an economic loss to the farmers, ammonia volatilization has a deleterious effect on the environment and human health [3].

*N*-(*n*-butyl) thiophosphoric triamide (NBPT) is a urease inhibitor that has been widely reported to decrease ammonia volatilization from urea-based fertilizers under different soil and environmental conditions [2,4,5]. The NBPT is usually used to coat urea granules or mixed with liquid urea-based fertilizers, such as urea ammonium nitrate. The NBPT reduces ammonia volatilization by suppressing the activity of urease enzymes responsible for the hydrolysis of urea [6]. To suppress urease activity, NBPT becomes converted to either *N*-(*n*-butyl) thiophosphoric diamide (NBPD) or *N*-(*n*-butyl) phosphoric triamide (NBPTO) in soils [1,7]. The NBPD and NBPTO become hydrolyzed into monoamido thiophosphoric acid and diamido phosphoric acid, respectively. These then block the two nickel ions' active sites of urease enzymes via two oxygen atoms and one amide group [1]. The global

efficiency of NBPT in reducing ammonia volatilization from urea relative to untreated urea has been estimated to be 52% [2].

Several studies have reported a reduction in the efficiency of NBPT when combined with a nitrification inhibitor (NI) in reducing ammonia volatilization from urea [8–13]. The decrease in NBPT efficiency when used as NBPT plus NI (double inhibitor, DI) was attributed to the persistence of ammonium by the NI. Recent studies to elucidate the mechanism of the reduced NBPT efficacy in decreasing ammonia volatilization with DI showed that NI, specifically 3,4-dimethyl pyrazole phosphate, significantly impaired the inhibitory effect of NBPT on urea hydrolysis rates across several soils and temperatures [14,15]. For example, NI was found to reduce the inhibitory effect of NBPT on the urea hydrolysis rate by 21% across five soils at 21 °C [14]. The effectiveness of NBPT in reducing the urea hydrolysis rate decreases as temperature increases [15]. The action of NBPT on urea has been shown to be more effective in reducing ammonia volatilization during fall than spring seasons on the Canadian prairies [13]. Moreover, studies have also shown that the rate of NBPT degradation in soils was greater in acidic than alkaline soils [16,17]. However, there is a lack of information on the influence of NI on the degradation of NBPT with or without urea in soils. This study was conducted to evaluate the influence of NI, particularly 3,4-dimethyl pyrazole phosphate, on the degradation rate of NBPT without urea in six soils. We hypothesized that NI would interfere with the persistence of NBPT in soils.

## 2. Materials and Methods

### 2.1. Soil Characteristics

An incubation study was conducted on soils (0–15 cm depth) that were collected from six locations in Manitoba, Canada. The locations were Carman (Soil 1; 49°29′6″N, 98°02′2″W), Carberry (Soil 2; 49°53′7″N, 99°22′29″W), Deerwood (Soil 3; 49°22′1″N, 98°23′34″W), High Bluff (Soil 4; 50°01′ 2″N, 98°08′9″W), Beausejour (Soil 5; 50°05′13″N, 96°29′58″W), and Portage la prairie (Soil 6; 49°57′9″N, 98°16′0″W). These were the same six soils used in two previous studies [14,15]. In the Canadian soil classification system, all soils are classified as Chernozems (an equivalent of Chernozem in the FAO classification system) except Soil 4, which is classified as a Regosol (an equivalent of Regosol in the FAO classification system) [18]. The soils were air-dried and ground to pass through a 2-mm sieve. A subsample of each soil was analyzed (Table 1) for organic matter by the wet oxidation method [19], cation exchange capacity by ammonium acetate method [20], urease activity [21], soil texture by pipette method [22], field capacity [23], and pH (soil/water, 1:2) and electrical conductivity with a combined conductivity and pH meter (Orion versaStar, ThermoFisher Scientific Inc., Waltham, MA, USA).

### 2.2. Experimental Design and Treatment Applications

The experimental setup was a completely randomized design containing two inhibitor treatments of six soils, a factorial layout for eight sampling periods, and was replicated three times for a total of 288 experimental units. The inhibitor treatments were NBPT (10 mg NBPT $kg^{-1}$ soil) and NBPT plus NI (DI; 10 mg NBPT + 2.5 mg NI $kg^{-1}$ soil). We used analytical grades of NBPT (CAS: 94317-64-3) and NI (3,4-dimethyl pyrazole phosphate; CAS: 202842-98-6) in this study.

Ten grams of each soil was weighed in 50 mL centrifuge tubes. The soil was wetted to 75% field capacity based on soil mass, capped, and left to equilibrate for 24 h at room temperature. Twenty-four hours after wetting, the soils in the centrifuge tubes were spiked with 0.5 mL of a solution containing either 200 mg NBPT $L^{-1}$ (NBPT inhibitor treatment) or 200 mg NBPT + 50 mg NI $L^{-1}$ (DI inhibitor treatment). The ratio of NBPT to NI in the DI inhibitor treatment was the same as the ratio of NBPT to NI in the double inhibitor formulation used in our previous studies [14,15]. However, the current study did not include urea with the inhibitors, as we discovered that the presence of urea interfered with the analytical procedure for NBPT. The tubes were recapped and placed in an incubator (Isotope Incubator, Model 304, Fisher Scientific, Hampton, NH, USA) set at 21 °C. On days

0, 0.5, 1, 2, 4, 7, 10, and 14 after treatment application, three replicates or samples of each soil by inhibitor treatment (i.e., six soils × two inhibitor treatments × three replicates for a total of 36 samples) were removed (destructive sampling) from the incubator for NBPT extraction and analysis. Day 0 was immediately after the soil was spiked with the inhibitor treatments.

**Table 1.** Selected soil (0–15 cm) properties.

| Soil Property | Soil 1 | Soil 2 | Soil 3 | Soil 4 | Soil 5 | Soil 6 |
|---|---|---|---|---|---|---|
| Soil classification [a] | Orthic Black Chernozem | Orthic Black Chernozem | Orthic Dark Gray Chernozem | Gleyed Cumulic Regosol | Gleyed Rego Black Chernozem | Gleyed Rego Black Chernozem |
| Soil series | Hibsin | Fairland | Dezwood | High Bluff | Dencross | Neurhorst |
| Soil $pH_{water}$ | 5.51 | 6.65 | 6.62 | 7.46 | 7.76 | 7.96 |
| Electrical conductivity ($\mu S\ cm^{-1}$) | 394 | 228 | 1853 | 899 | 1377 | 596 |
| Organic matter ($g\ kg^{-1}$) | 27 | 33 | 34 | 45 | 88 | 71 |
| Available N ($mg\ kg^{-1}$) | 31 | 15 | 186 | 58 | 22 | 82 |
| Field capacity ($m\ m^{-3}$) | 0.35 | 0.34 | 0.36 | 0.41 | 0.61 | 0.44 |
| Urease activity ($mg\ NH_4^+\text{-}N\ kg^{-1}\ soil\ hr^{-1}$) | 11 | 17 | 24 | 57 | 63 | 88 |
| Cation exchange capacity ($cmol\ kg^{-1}$) | 16 | 14 | 23 | 28 | 47 | 36 |
| Soil texture | Sandy loam | Sandy loam | Loam | Loam | Clay | Clay loam |
| Sand ($g\ kg^{-1}$) | 711 | 764 | 465 | 427 | 108 | 269 |
| Silt ($g\ kg^{-1}$) | 123 | 128 | 318 | 325 | 322 | 343 |
| Clay ($g\ kg^{-1}$) | 166 | 108 | 217 | 248 | 570 | 388 |

[a] Canadian soil classification system.

### 2.3. Extraction and Analysis of NBPT

On each sampling day, 25 mL of deionized water was dispensed on the sampled centrifuge tubes and shaken on a reciprocating shaker for 30 min at 120 excursions per minute. After 30 min of shaking, the samples were centrifuged for 5 min at $10,000\times g$ to allow soil residues to settle to the bottom. Immediately after centrifugation, about 4 mL aliquot was transferred using a 0.2 μm syringe filter (Basix™ Syringe Filters, ThermoFisher Scientific, Waltham, MA, USA) into a 20 mL vial. This was followed by transferring 1 mL of the filtered aliquot into a 2 mL high-performance liquid chromatography (HPLC) vial (9 mm surestop screw vial, ThermoFisher Scientific, Waltham, MA, USA) containing 0.1 mL dimethyl sulfoxide for NBPT analysis with HPLC-mass spectrometry (HPLC-MS), as described by Engel et al. [17].

The HPLC-MS (Bruker Compact QqTOF, Billerica, MA, USA) used was equipped with an electrospray source that operated in the positive ionization mode. The nebulizer pressure of the source was 0.3 bar with 5 L $min^{-1}$ of $N_2$ drying gas at 200 °C. The capillary voltage was 3500 V, and the capillary exit voltage was 70 V. Reverse-phase chromatography was used to separate NBPT using an Intensity Solo C18 (100 × 2.1 mm, 2 μm) HPLC column (Bruker Daltonik, Billerica, MA, USA). The column was maintained at 35 °C with a flow rate of 300 μL $min^{-1}$. The mobile phase consisted of formic acid 0.1% in Milli-Q water for Channel "A" and acetonitrile for Channel "B". A 2 μL aliquot of the sample was injected into the column and kept at 80% B from 0 to 3 min. From 3 to 4 min, the gradient was linearly ramped to 20% B, where it was kept for 1.5 min. Then, the gradient was linearly ramped to 80%, and it was held for 2.5 min at 80% for re-equilibration. The NBPT was eluted at approximately 3.3 min.

Data quantitation was performed using Bruker Daltonic QuantAnalysis (ver. 4.4) software (Billerica, MA, USA). The ion chromatograms for NBPT were defined as $[M + H]^+$ (168.0719 *m/z*). The concentrations of NBPT recovered in soil were determined from a calibration curve of known standard solutions of NBPT and their corresponding peak

areas. The quantity of NBPT recovered was expressed as a percentage of NBPT applied to the soils.

*2.4. Kinetics and Statistical Analysis*

Model fitting and statistical analysis were performed with SAS software (SAS Institute 2014, ver. 9.4 [24]). PROC NLIN was used to fit an exponential decay function (Equation (1)) to determine the degradation rate constant (k) of NBPT in the soils as follows:

$$Y = b_o[\exp(-kt)] \tag{1}$$

where Y is the % of NBPT recovered in soils at time t, t is the time in days, k is the NBPT degradation rate constant, and $b_o$ is an empirical constant.

For ease of interpretation, the generated k was used to calculate the half-life ($t_{1/2}$) of NBPT in the inhibitor treatments using Equation (2):

$$t_{1/2} = \ln(2)/k \tag{2}$$

We used PROC GLIMMIX (beta distribution) for repeated measure analysis to determine the significant effect of time, inhibitor treatments, and their interaction on the % of NBPT recovered in each soil. Furthermore, analysis of variance with PROC GLIMMIX (gamma distribution) was performed on the degradation rate constant and half-life of the NBPT across soils and inhibitor treatments. The fixed effects in the model were soil and inhibitor treatment. Mean comparisons were deemed significant at a probability level of 0.05 Fishers' protected least-significant difference. The goodness of fit for the exponential decay model was tested using the Nash–Sutcliffe model efficiency (ME) and root means square error [25]. Stepwise regression with PROC REG was used to analyze the influence of soil properties on the half-life of NBPT in soils.

$$ME = 1 - \frac{\sum_{i=1}^{n} \left(Y_i^m - Y_i^p\right)^2}{\sum_{i=1}^{n} \left(Y_i^m - \overline{Y}\right)^2} \tag{3}$$

where $Y_i^m$ is the measured NBPT recovered in soil, $Y_i^p$ is the predicted NBPT recovered in soil, and $\overline{Y}$ is the mean of measured NBPT recovered in soil. When *ME* = 1, there is a perfect relationship measured and predicted NBPT recovery in soil; and when *ME* = 0, the model has the same precision as the mean of measured NBPT recovered.

## 3. Results

*3.1. NBPT Recovery*

The interaction of treatment and time did not significantly affect % NBPT recovered, except in Soil 2 (Table 2). The significant interaction in Soil 2 was because of greater NBPT recovered in NBPT only with DI on 0.5 and 2 d (Figure 1). The % of NBPT recovered immediately after treatment application (time = 0 d) was less than 70% in all the soils except in Soil 1 (Figure 1). The low recovery of NBPT on 0 d might be due to other NBPT species (e.g., [M + Na]$^+$) of the ion chromatograms that were not accounted for. As expected, the % of NBPT recovered significantly decreased with time in an exponential decay order (Figure 1). The persistence of NBPT was shortest in Soil 1, with the NBPT recovery reaching the lowest point (3%) by 2 d in both inhibitor treatments. In contrast, the % of NBPT recovered from neutral to alkaline soils on 2 d ranged from 22 to 44% in both inhibitor treatments (Figure 1). By 7 d, NBPT was below the detection limit in all soils. The NBPT recovery was well predicted by the exponential decay function as indicated by the Nash–Sutcliffe model efficiency, which ranged from 0.93 to 0.99 across the soils.

**Table 2.** Effect of inhibitor treatment and time on % of NBPT recovered in soils.

| Model Effect | df | Soil 1 | Soil 2 | Soil 3 | Soil 4 | Soil 5 | Soil 6 |
|---|---|---|---|---|---|---|---|
| | | Probability values | | | | | |
| Inhibitor treatment (I) | 1 | 0.8588 | 0.8945 | 0.8266 | 0.813 | 0.9467 | 0.9731 |
| time (t) | 7 | <0.0001 | <0.0001 | <0.0001 | <0.0001 | <0.0001 | <0.0001 |
| I × t | 7 | 1.0000 | 0.0199 | 0.3853 | 0.8640 | 0.4941 | 0.7716 |

Probability values are significant at <0.05. df, degree of freedom.

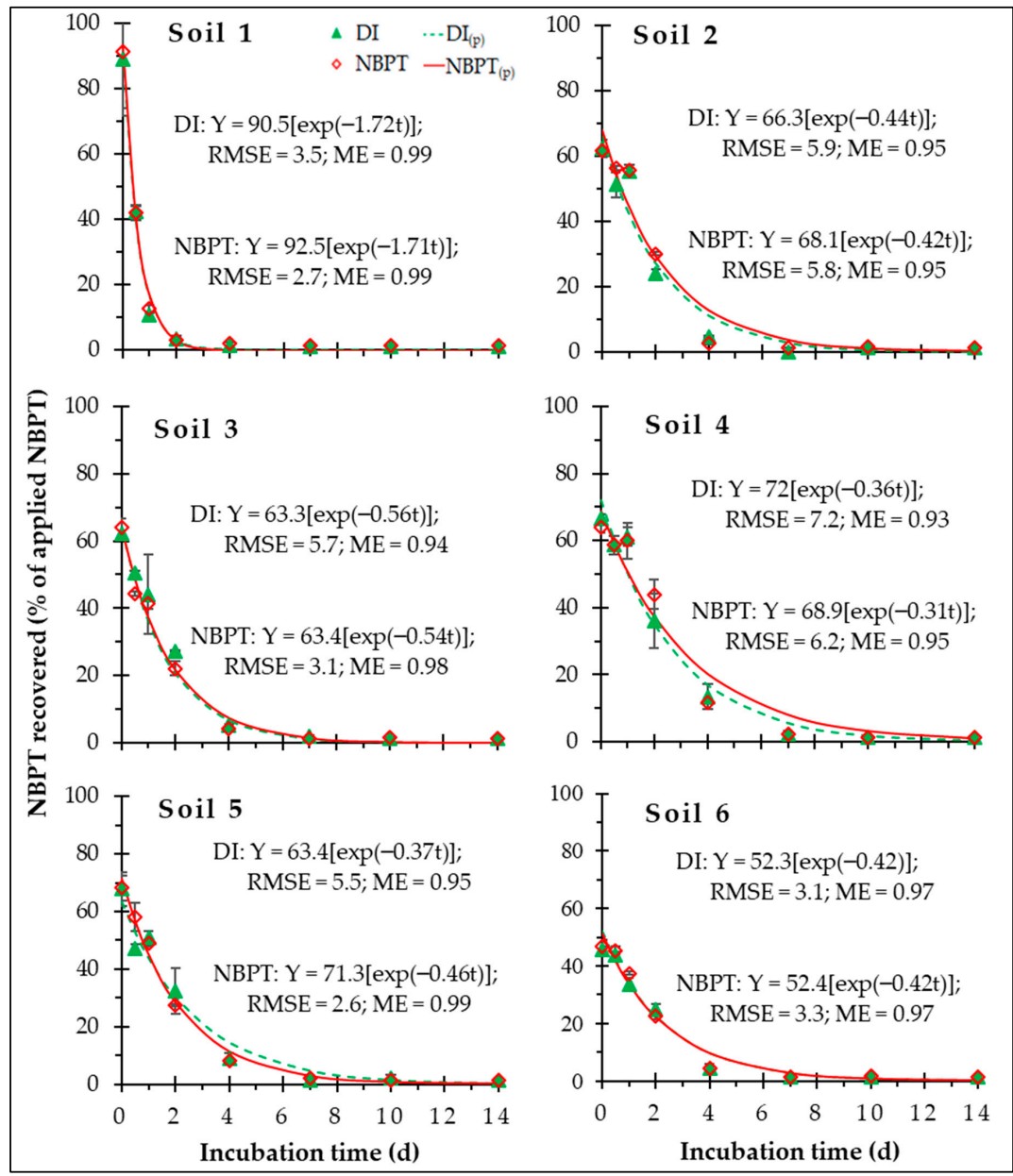

**Figure 1.** The percentage recovery of NBPT in soils. Error bars are standard errors of the mean (*n* = 3). NBPT and NBPT(p) are measured and predicted % of NBPT recovered in NBPT inhibitor treatment; DI and DI(p) are measured and predicted % of NBPT recovered in DI inhibitor treatment. *N*-(*n*-butyl) thiophosphoric triamide; DI, double inhibitor [NBPT + 3,4-dimethyl pyrazole phosphate]; Y, % of applied NBPT recovered; RMSE, Root mean square error, ME, Nash–Sutcliffe model efficiency.

### 3.2. Kinetics of NBPT Degradation

The NBPT degradation rate constant was not significantly affected by inhibitor treatment or the interaction of soil and inhibitor treatment (Table 3). As such, the half-life of NBPT in each soil was not affected by the type of inhibitor treatment (NBPT only versus DI; Table 3). Averaged across soils, the half-life of NBPT in either inhibitor treatment was 1.3 d. The lack of a significant difference in the half-life of NBPT between the two inhibitor treatments did not agree with our hypothesis. Our previous study had found that NI reduced the half-life of NBPT-treated urea in soils by 1 d at 21 °C [15].

**Table 3.** Effect of inhibitor treatment and soil on degradation rate constant (k) and half-life ($t_{1/2}$) of NBPT.

| Group Means | k | $t_{1/2}$ |
|---|---|---|
| Inhibitor treatment (I) | d | $d^{-1}$ |
| NBPT | 0.54 a | 1.32 a |
| DI | 0.54 a | 1.33 a |
| Soil (S) | | |
| Soil 1 | 1.72 a | 0.44 d |
| Soil 2 | 0.43 c | 1.61 b |
| Soil 3 | 0.55 b | 1.30 c |
| Soil 4 | 0.33 d | 2.09 a |
| Soil 5 | 0.41 c | 1.71 b |
| Soil 6 | 0.42 c | 1.66 b |
| Model effects | Probability values | |
| I | 0.8784 | 0.8876 |
| S | <0.0001 | <0.0001 |
| I × S | 0.5452 | 0.5021 |

Note. Means with different letters within a column are significantly different at a probability value of <0.05 using Fisher protected LSD.

Unlike the inhibitor treatment, there was a significant effect of soil on the half-life of NBPT in soils. Soil 1, which was the acidic soil, had the shortest half-life (0.4 d), while Soil 4, which was slightly alkaline, had the longest half-life (2.1 d) when averaged across inhibitor treatments (Table 3). NBPT is *N*-(*n*-butyl) thiophosphoric triamide, DI is double inhibitor [*N*-(*n*-butyl) thiophosphoric triamide + 3,4-dimethyl pyrazole phosphate]. The shortest half-life of NBPT in Soil 1 was consistent with previous studies that found that the NBPT degradation rate was faster in acidic than alkaline soils [16,17]. Similarly, the shortest half-life of the inhibitor treatments in Soil 1 corroborated our previous studies that used the same soils and found the half-life of urea treated with either NBPT or DI to be shorter in Soil 1 than in other soils [14,15]. Additionally, other studies had also reported a lower NBPT inhibition of urea hydrolysis in acidic than alkaline soils [26,27]. The lack of NI on NBPT degradation in this study was probably because of the absence of urea. This is because the soil pH around applied urea changes during the hydrolysis of urea and the nitrification process. This implies that NI did not affect the persistence of NBPT in soil but rather impaired the inhibitory effect of NBPT on urea hydrolysis. With no effect of NI on NBPT degradation, the observed inhibition of NBPT to reduce urea hydrolysis by NI, as noted in the studies [14,15] might have been because of the soil acidification during nitrification [28]. While hydrolysis of NBPT in soils to form NBPTO and NBPD is required to inhibit the process of urea hydrolysis [1,7], rapid hydrolysis of NBPT, as shown in the case of acidic soil, may be counter-effective. For example, NBPTO and phenyl phosphorodiamidate are potent urease inhibitors with greater inhibition of urease than NBPT under a buffered solution, but their reduced persistence in soils makes them less effective in reducing ammonia volatilization when compared to NBPT [16,29–31].

Stepwise regression analysis showed that soil pH, organic matter, and urease activities accounted for 91% of the variation in the half-life of NBPT in soil. Of these soil properties, soil pH was the most predictive factor of NBPT half-life, as indicated in Equation (4).

The persistence of NBPT in soils increased as the soil pH increased from strongly acidic to neutral soil pH and then decreased from neutral soil pH to slightly to moderately alkaline soil pH (Figure 2; pH classification based on USDA). An earlier study had shown that the half-life of NBPT in acidic soil (pH = 4.9) could be extended by 2.5 d when the soil pH was increased to neutral pH (6.9) using calcium hydroxide [16]. Despite the reported reduced persistence of NBPT in acidic than alkaline soils, the reduction of ammonia volatilization by NBPT relative to untreated urea is not always lower in acidic than alkaline soils [32].

$$\text{Half-life} = -5.874 + 1.221(\text{pH}) - 0.0141(\text{urease activity}) - 0.1163(\text{organic matter}) \quad (4)$$

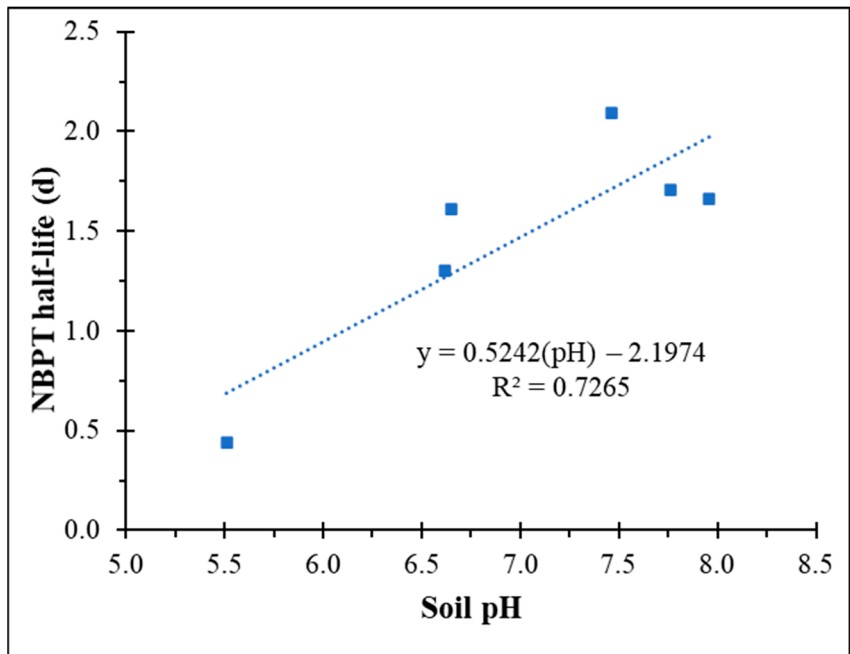

**Figure 2.** Relationship between soil pH and NBPT half-life. Y is NBPT half-life in soil.

### 4. Conclusions

The urease inhibitor, NBPT, plays an important role in conserving applied urea-N in the soil. While NI is known to impair the inhibitory effect of NBPT on urea hydrolysis, our study showed that NI did not interfere with the persistence of NBPT in soil. Instead, the persistence of NBPT in soil was mainly influenced by soil pH. We found that the degradation of NBPT was two to four times greater in acidic than neutral to alkaline soils. The half-life of NBPT was 0.4 d in acidic soil and 1.3 to 2.1 d in neutral to alkaline soils. As such, N management with NBPT may be more suitable for alkaline than acidic soils, and alkaline soils thereby provide more flexibility in precipitation or irrigation scheduling to incorporate urea into the soil while reducing N losses. Future studies will need to evaluate how the interaction between urea, NI, and NBPT affect the persistence of NBPT over a wide range of soils and environmental conditions.

**Author Contributions:** Conceptualization, A.A.L.; Data curation, A.A.L.; Formal analysis, A.A.L.; Funding acquisition, O.O.A.; Investigation, A.A.L.; Methodology, A.A.L.; Project administration, O.O.A.; Supervision, O.O.A.; Validation, A.A.L.; Visualization, A.A.L.; Writing—original draft, A.A.L.; Writing—review and editing, A.A.L. and O.O.A. All authors have read and agreed to the published version of the manuscript.

**Funding:** This research received no external funding.

**Data Availability Statement:** Data available upon request.

**Acknowledgments:** The authors appreciate the technical support of Emily Komatsu in method development for measuring NBPT with the HPLC-MS.

**Conflicts of Interest:** The authors declare no conflict of interest.

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
