# Peer review of "Degradation of N-(n-butyl) Thiophosphoric Triamide (NBPT) with and without Nitrification Inhibitor in Soils"

_nitrogen, doi:10.3390/nitrogen3020012_

Round 1
Reviewer 1 Report
Dear authors the manuscript entitled: 'Degradation of N-(n-butyl) thiophosphoric triamide (NBPT) 2 with and without nitrification inhibitor in soils' is well presented. English needs minor revision.
The issue addressed by the submitted manuscript is the influence of 3,4- dimethyl pyrazole phosphate (a Nitrification inhibitor) on the degradation of N-(n-butyl) thiophosphoric triamide (NBPT) in six soils with different properties. The work encompasses components of scientific novelty.
As regards the NBPT recovery presented in figure 1 (one figure/soil) along with error bars are standard errors of the mean and explained in paragraph 3.1 needs clarifications as regards the number of measurements performed per single point. Also, clarifications needed on the number of times performed each experiment per soil type. One single experiment per soil type is not enough to reach a conclusion.
The half-life of NBPT has been considered based on one series of experiment? If this is the case a consistent conclusion could not be reached.
The authors concluded that the degradation of NBPT was 2 to 4 times greater in acidic than neutral to alkaline soils. How many acidic and alkaline soils have been tested?
What are the environmental conditions affect the NBPT degradation?
Please add in the manuscript the validation data of the LC-MS method used for the determination of N-(n-butyl) thiophosphoric triamide (NBPT), eg linearity, recovery etc.
Reviewer 2 Report
In my opinion the paper contains elements of scientific novelty, deals with an interesting and valuable experiment. I would recommend its' publication after minor revisions having been addressed :
Line 54, please add one line of the comment on climate impact ( meaning the climate zone in the simplest manner) on the reduced NBPT efficacy in decreasing ammonia volatilization.
Line 131- would you please write if it is a typical/default function to be used and why?
Line 139 - if possible, a reference is needed to better stress the meaning.
Results- 3.1 NBPT Recovery - A question: How many measurement repetitions on a single soil were performed? Figure 1 shows one series for each soil. Were there more of them on a single sample? Is is the case? As far as I am concerned, the samples had been properly prepared, by reciprocation and so on, adding proper substances /additives to perform chromatography analyses on a solution. How the way of preparing the sample affects the results? A number of sampels to draw the standard error?
Figure 2 - It is a reliable relationship only for 6 measurement points?
Line 238 - a practical conclusion would be nice , referring to implications for soil management/fertilization?
Round 2
Reviewer 1 Report
Authors should avoid so many self-citations.